# Continuous emergence of phototaxis in *Dictyostelium discoideum*

Damien Genettais[1], Charles Bernard[2], Félix Geoffroy[3], Clément Nizak[4]*, Sandrine Adiba[1]*

1 Institut de Biologie de l'ENS (IBENS), Département de biologie, Ecole normale supérieure, CNRS, INSERM, Université PSL, Paris, France, 2 Institut Pasteur, Université de Paris, CNRS UMR3525, Microbial Evolutionary Genomics, Paris, France, 3 Société Nationale de Protection de la Nature, Paris, France, 4 Sorbonne Université, CNRS, Institut de Biologie Paris-Seine, Laboratoire Jean Perrin, Paris, France

* clement.nizak@sorbonne-universite.fr (CN); sandrine.adiba@ijm.fr (SA)

**Data availability statement:** All the data used in this manuscript and the code necessary to reproduce all the figures from this dataset is available in the Zenodo repository: https://doi.org/10.5281/zenodo.13245281.

## Abstract

The evolutionary transition from uni- to multicellularity is associated with new properties resulting from collective cell behavior. The social amoeba *Dictyostelium discoideum* alternating between individual cells and multicellular forms of varying size provides a powerful biological system to characterize such emergent properties. Multicellular forms coined slugs have long been described as chemotactic towards cAMP, and also as phototactic. While chemotaxis is also well-documented at the single-cell level, which explains slug chemotaxis, we asked whether slug phototaxis is an emergent property of multicellularity. For this, we developed an automated microscopy setup to quantify and compare the migration trajectories of single cells and slugs moving in the dark or illuminated with lateral light. We find that single cells, either extracted from phototactic slugs or taken prior to multicellular aggregation, are not phototactic, implying that slug phototaxis results from interactions between cells that lack this property. Further, by analysing slugs composed of a varying number of cells, we find that phototaxis efficiency increases continuously with slug size. Cell-cell interactions combined with self-organization are thus key elements for this property to emerge.

## Introduction

Collective behaviors are widespread in biological systems at various scales [1] ranging from molecules [2], bacterial colonies [3] to groups of individuals such as insects [4], fish schools [5], or bird flocks [6]. What these diverse systems hold in common is the process of self-organization arising from local interactions between lower-level components through coordination of individual actions. The spontaneous emergence of this spatio-temporal order may give rise to new properties of the system that their single elements do not possess. These emergent properties have fundamental effects on the group as a whole, as well as on the individuals within it. Indeed, this coordination can modify the activity patterns of individuals, namely increasing foraging [7,8], enhancing reproductive success [9] or improving safety [10].

**Funding:** This work has received support under the program "Investissements d'Avenir" launched by the French Government and implemented by ANR with the references ANR–10–LABX–54 MEMOLIFE and ANR–10–IDEX–0001–02 PSL* Université Paris, Q-life ANR-17-CONV-6150005.

**Competing interests:** The authors have declared that no competing interests exist.

Probing experimentally and quantitatively how such a collective property emerges as the number of interacting individuals increases is both valuable and challenging, and has remained scarce. In a recent study analysing colonies of 10 to 200 ants, response to overheating was shown to depend on a group size-dependent threshold [11]. Here we analysed the phototactic behavior of the social amoeba *Dictyostelium discoideum* to demonstrate its emergent character and characterize how phototaxis efficiency depends on social group size from 354 to $5.3 \times 10^4$ cells.

*D. discoideum* social amoebae alternate between single vegetative cells feeding on bacteria and multicellular forms upon nutrient starvation, providing a unique opportunity to compare behaviors at two organizational levels in the same organism. Populations of solitary cells may reach $10^7$ to $10^9$ cells. When food is exhausted, individual cells aggregate through cyclic Adenosyl MonoPhosphate (cAMP) signalling (chemotaxis) to form multicellular slugs composed of typically $10^2$ to $10^5$ cells. Slugs can cross a layer of soil to reach the surface by detecting light via their tip (prestalk region) and turn in less than 5 minutes upon light source switching [12]. After migration, slugs differentiate into spore-containing fruiting bodies. *D. discoideum* slugs are chemotactic towards cAMP, merely as a result of the same property already present at the single cell level. On the other hand, phototaxis, that has been widely reported in multicellular slugs, has led to contrasting observations concerning the response of isolated cells. Some studies [13–15] report no evidence of phototaxis in single cells, while others [16–18] show cell accumulation or dispersion near a light source. These discrepancies may stem from the different experimental methodologies used in these studies. In addition, these methodologies characterized cell or slug motion mainly indirectly, e.g. via the analysis of traces left behind by cells or slugs on agar. To revisit this question, we have thus developed an approach for the direct visualization and statistical analysis of single-cell and slug motion in controlled lateral illumination conditions.

We first investigated whether phototaxis is an emergent property of multicellularity by subjecting slugs, vegetative cells and cells from dis-aggregated phototactic slugs to lateral light or to dark conditions and tracked their trajectories. Statistical analysis of the migration of single cells vs slugs, either in the dark or illuminated laterally, revealed that single vegetative cells and cells from dis-aggregated phototactic slugs displayed no detectable phototaxis, compared to slugs that migrated towards light, demonstrating that phototaxis is an emergent property of multicellularity.

To characterize this emergent behaviour, we then quantified the slugs' response to light as a function of their cell number (depending on plated cell density and thus experimentally tunable). We have developed an automated time-lapse microscopy and image analysis pipeline at suitable spatio-temporal resolution and scale to quantify a large number of slug trajectories (661 slugs when exposed to light and 593 in the dark) at the macro-scale and to extract morphological data and slug cell number at the micro-scale. By quantifying phototaxis efficiency and slug migration over a wide range of slug sizes, we observed a gradual increase of collective phototactic efficiency as a function of slug cell number.

## Materials and methods

### Strains and culture conditions

The amoeba *Dictyostelium discoideum* HM388 (Dictybase ID: : DBS0236290), an axenic strain was used for all phototaxis experiments. This strain is a slugger mutant, carrying a deletion in the *mybC* gene. As a result, this strain exhibits an extended slug migration time lasting up

to 60 hours. The *mybC* mutation was reported not to impact the organization of the multi-cellular slug but instead to affect culmination, a later stage of the developmental cycle [19]. Qualitatively identical observations of slug size-dependent phototaxis were obtained with the parent wild-type strain, which was less convenient to accumulate large datasets (typical slug migration time of 4h-6h). In order to estimate the number of cells within slugs, this strain was transformed with autonomous extra-chromosomal plasmids pTX-GFP (Dictybase ID: 11) or pTX-RFP (Dictybase ID: 112) to express either GFP or RFP fluorescent markers respectively. The fluorescent proteins encoded on the plasmid also carries a gene for antibiotic resistance (Gentamicin 418: G418, Sigma-Aldrich). Cells were cultured in autoclaved HL5 medium (per L, 35.5 g HL5 from formedium, pH = 6.7) at 22°C. A concentration of 20 µg mL$^{-1}$ G418 was added when transformed cells were cultured. All experiments were run in a dark room at a room temperature of 21 ± 1°C, regulated by the air conditioning system of the laboratory.

## HM388 transformation

Cells were transformed as in Adiba et al. [20] using a standard electroporation procedure with pTX-GFP or pTX-RFP. HM388 cells were grown in 75 cm$^2$ flasks until they reached high cell density (but before stationary phase of their growth). Four to six hours before the transformation, fresh medium was added. Cells were then re-suspended in 10 mL of ice-cold HL5 and kept on ice for 30 min. Cells were centrifuged for 5 min, 500 g at 4°C. The pellet was then re-suspended in 800 µL of electroporation buffer and transferred into ice-cold 4 mm electroporation cuvettes containing 30 µg of plasmid DNA. Cells were electroporated at 0.85 kV and 25 mF twice, waiting for 5 seconds between pulses and transferred from the cuvette to 75 cm$^2$ flask with fresh HL5 medium. The antibiotic G418 at 5 µg mL$^{-1}$ was added to the culture media the next day to select for transformants. The concentration of G418 was then gradually increased from 5 µg mL$^{-1}$ to 20 µg mL$^{-1}$ over 1–2 weeks and resistant cells were collected and frozen.

## Multicellular development

Cells from mid-logarithmic cultures were centrifuged (500 g; 7 min) and washed three times with SorC buffer (per L, 0.0555 g CaCl$_2$; 0.55 g Na$_2$HPO$_4$·7H$_2$O; 2 g KH$_2$PO$_4$). For all phototaxis experiments, a volume of 40 µL was plated on Petri dishes with 2% phytagel (as described in Dubravcic et al. [21]), wrapped with aluminum foil (to avoid light leaking from the outside) until the phototaxis assay (24 hours after plating). This time of incubation was sufficient for the leading slugs formed from the amoebae at the origin to migrate near the edge of the spot of plated cells.

## Phototaxis assay

**Individual cell phototaxis**   Individual cell phototaxis was performed for vegetative cells and cells obtained from dis-aggregated slugs. Before dis-aggregation, slugs were illuminated as described in the section 'slug phototaxis' (light condition) during 6 hours. Slugs were then picked up from the plates and placed in 1 mL tube containing 500 µL SorC, and dis-aggregated by pipetting the slug suspension. Vegetative cells or cells from dis-aggregated slugs were plated at low density (10$^4$ cells/mL). After 30 min—the time for the cells to attach to the bottom of the dish—cells were imaged as in dark and light conditions described in the section 'slug phototaxis'. An area of the Petri dish was scanned to analyse 200 to 300 cell trajectories for three biological replicates (see 'Time lapse image acquisition' section).

To disentangle between self-organization and cell-cell interaction during phototaxis migration, vegetative cells at high densities ($5.10^7$ cells/mL, 3 replicates) were plated and individual cell trajectories tracked as before. To do so, a low percentage of HM388 RFP vegetative cells (around 0.1%) was mixed with HM388 GFP vegetative cells, allowing individual RFP cells to be tracked. Before plating, cells were washed once with HL5 to avoid starvation and then self-organization emergence.

**Slug phototaxis** Cells were starved as described in the section 'Multicellular development' without light during 24 hours before starting phototaxis assays. For experiments in the dark, cells were inoculated into the centre of the Petri dish. For slugs exposed to light, cells were placed at 1 cm from the periphery of the Petri dish and light source located at the opposite side. During image acquisition, the Petri dish was covered with a black ring made with a 3D printer for 'dark' condition. For lateral light condition, a black holed ring (see S1_Fig A) allowed a white light LED (450 nm maximum wavelength) to pass through a 2 mm diameter hole and irradiated the phytagel surface on the opposite side of slugs formed. The LED light source eliminates possible heat effects (thermotaxis). To obtain a large range of slug length, 3 to 4 replicates with 2 sub-replicates were performed using cell densities ranging from $1.5 \times 10^6$ to $1.6 \times 10^7$ cells/cm$^2$ (see S1_Table).

**Fluorescent chimeric slugs** To estimate the number of cells within slugs, 1% HM388 GFP cells were mixed with 99% HM388 RFP cells after three centrifugation of each strain with SorC. We also performed the reciprocal mix of 1% HM388 RFP with 99% HM388 GFP to control for fluorescent reporter biases. The percentage of fluorescent cells within the mix was measured using a cytometer (FACS Cube8) to obtain an accurate quantification of this fraction. A volume of 40 µL of the cell mix was plated on 6 cm Petri dishes containing 2% phytagel and was then exposed to light or in the dark as in our 'slug phototaxis' assay.

## Time lapse image acquisition

The 6 cm diameter Petri dish was imaged using a 5× objective and an automated inverted microscope Zeiss Axio Observer Z1 with a Camera Orca Flash 4.0 LT Hamamatsu. Images were acquired with MicroManager 1.4 software. The Petri dish was scanned at regular time intervals (typically 10 min for slug phototaxis and 30 seconds for individual cell phototaxis), with phase contrast image acquisition (33 ms exposure times and with the lowest light illumination) and during 50 hours or 2 hours for slug and individual cell phototaxis respectively. To estimate the number of cells within slugs, another assay was performed adding fluorescence image acquisition during 5.5 hours. Three replicates mixing 1% RFP and 99% GFP HM388 cells were subjected to light and dark conditions. To control for the effect of the fluorescent marker, a mix of 1%GFP and 99% RFP HM388 cells (3 replicates) was also analysed in the dark.

## Data analysis

**Individual cell tracking** Individual cell trajectories were automatically extracted from time lapse movies using the Python package Trackpy [22]. *Center trajectories* and *Polar angle* were obtained by computing Eqs (1) and (2) from (x,y) cell's coordinates.

**Center trajectories** *Center trajectories* were plotted after computing differences between x, y barycenter's coordinates at time (t=t) and at the starting time (t=0) :

$$\begin{aligned} \mathrm{X}_c &= x(t) - x(0) \\ \mathrm{Y}_c &= y(t) - y(0) \end{aligned}$$

(1)

where $x(t),y(t)$ [$\mu$m] : cell's barycenter coordinate at time $t$ and $x(0),y(0)$ [$\mu$m]: cell's barycenter coordinate at time 0.

**Start-end direction** *Start-end directions* were computed as the direction of the vector linking the start (t=0) and the end (t=T) of each slug's trajectory:

$$\theta_{\text{SE}} = \arctan\Big(x(0) - x(T), y(0) - y(T)\Big) \tag{2}$$

**Macroscale slug tracking and trajectory analysis** A custom Python program reconstructed a tiled image (Macroscale picture; S1_FigA-b) by combining all images of contiguous areas of the Petri dish acquired at the micro-scale (Microscale pictures S1_FigA-a), for each time point. Slug trajectories were automatically extracted from Macroscale time lapse movies using the wrMTrck plugin ImageJ [23] (extracting x,y slug's barycenter coordinates) in the Region Of Interest (ROI S1_FigA-b orange square) - where slugs migrated and removing the area of cell aggregation (S1_Fig red polygon). Regions of interest were adjusted for light and dark conditions as in S1_FigB.

*Center trajectories* and *Polar angles* were obtained by computing (Eqs 1) and (2) from (x,y) slug's coordinates (in mm for slugs instead of μm for cells).

**Instantaneous direction** The *instantaneous direction* $\theta(t)$ between two successive positions (typically every 10 minutes) was computed as follow with $\Delta t = 1$:

$$\theta(t) = \arctan\Big(x(t) - x(t + \Delta t), y(t) - y(t + \Delta t)\Big) \tag{3}$$

**Direction variability** To estimate *direction variability*, we first computed the instantaneous direction $\theta$(t) (rad) and then estimated the variance of the direction over the slug trajectory ($rad^2$):

$$direction\ variability = variance(\theta(t)) \tag{4}$$

**Phototaxis efficiency ($\kappa$)** Phototaxis efficiency was defined using statistics of directional data from K.V. Mardia [24]. Directional data were fitted with the von Mises distribution (Eq 5), which yields a quantitative concentration parameter *kappa* ($\kappa$) describing the extent to which individual directions are clustered around an average direction. A value close to zero reflects a wide/uniform distribution of directions of slug migration whereas a large value reflects a perfect orientation; it is thus a measure of the orientation bias/preference of slug migration.

$$f(x, \kappa) = \frac{exp(\kappa.cos(x))}{2\pi I_0(\kappa)} \tag{5}$$

with $I_0(\kappa)$ the modified Bessel function of order zero:

$$I_0(\kappa) = \sum_{k=1}^{\infty} \frac{(\kappa^2/4)^k}{(k!)^2} \tag{6}$$

with $-\pi < x < \pi$

The parameter *kappa* was then estimated using the python library *scipy*.

**Velocity** The *velocity* (mm/hours) was computed along the slug trajectory and calculated from consecutive x,y coordinates of the slug's barycenter, according to the equation:

$$Velocity(t) = \frac{\sqrt{\Big(x(t + \Delta t) - x(t)\Big)^2 + \Big(y(t + \Delta t) - y(t)\Big)^2}}{\Delta t} \tag{7}$$

where $x(t + \Delta t), y(t + \Delta t)$ [mm]: slug's barycenter coordinate at time $t+\Delta t$, $x(t), y(t)$ [mm]: slug's barycenter coordinate at time $t$ and Velocity was then averaged for each slug along its trajectory.

*Velocity vector components parallel (y) and orthogonal (x) to the light source direction* (mm/h) were computed as follow:

$$V_x(t) = \left(x(t + \Delta t) - x(t)\right)/\Delta t$$
$$V_y(t) = \left(y(t + \Delta t) - y(t)\right)/\Delta t$$

(8)

where $V_x(t)$ : orthogonal instantaneous velocity component at time $t$ and $V_y(t)$ : parallel instantaneous velocity component at time $t$.

**Linear time**  The *linear time* was calculated by estimating the time during which the slug's *changing of direction*, relative to the start end direction, is lower than 10 degrees between two successive times, corresponding to a roughly straight line moving slug.

We first performed a rotation by the angle $\theta_{SE}$ of the landmark and computed the slug coordinates $(x_r, y_r)$ in this new landmark:

$$\begin{pmatrix} x_r(t) \\ y_r(t) \end{pmatrix} = \begin{pmatrix} cos(\theta_{SE}) & -sin(\theta_{SE}) \\ sin(\theta_{SE}) & cos(\theta_{SE}) \end{pmatrix} \cdot \begin{pmatrix} x(t) \\ y(t) \end{pmatrix}$$

Then the *changing of direction* $\theta_r(t)$ at each time step was computed as for *instantaneous direction* (Eq (3)) but using the slug coordinates $(x_r, y_r)$ in this new landmark with $\Delta t = 1$:

$$\theta_r(t) = \arctan\left(x_r(t) - x_r(t + \Delta t), y_r(t) - y_r(t + \Delta t)\right)$$

(9)

**Start end distance**  *Start en distance* of slug trajectories was computed using (x,y) coordinates at time zero and at the end of the migration (T) following equation:

$$SE = \sqrt{(x(T) - x(0))^2 + (y(T) - y(0))^2}$$

(10)

All of these parameters were computed as a function of the number of cells composing slugs. Ranges of cell numbers were obtained by subdividing data into six intervals such that each interval contains an equal number of slugs (up to 110 slugs).

**Microscale slug morphological analysis**

**Slug length**  Slugs were tracked using the wrMTrck plugin from ImageJ. Rows, columns and times where slugs were tracked within the macroscale tiled picture were extracted to analyse morphological features using microscale pictures (with 'measure region properties' from skiimage python library). A custom Python program allowed to avoid slugs that cross image borders, extracted main slug morphological features: area ($mm^2$), major and minor axis of ellipse fitted and perimeter (mm) and assigned the corresponding slug ID (SIμ) (S1_FigD) to the slug tracked with ImageJ at the macroscale ($SI_M$) at each time step. As moving slugs were not always elongated and might have a curvature when they migrated, slug length (mm) was calculated according to the equation:

$$length = \frac{perimeter}{2} - \frac{minoraxis}{2}$$

(11)

Length was then averaged for each slug along its trajectory.

**Number of cells within slugs**  The number of cells within each slug was assessed during migration either when in the dark or when exposed to lateral light. Slug migration was analysed at the macroscale as in the section 'Macroscale slug tracking and trajectories analysis'. Phase contrast and fluorescent images were then analysed with a custom python program to extract (1) slug length from phase contrast channel (as in the previous section) and (2) corresponding number of cells within each slug estimated from the fluorescent channel (S1_FigA-e) using local maxima detection from skiimage python library. The mean and standard deviation of slug length and number of local maxima were estimated along slug migration and a linear regression was then computed. To bound the lower limit of the number of cells to zero, a polynomial regression was performed for lower cell number (typically negative values of number of cells).

**Data pre-processing**  For all phototaxis analysis, data were pre-processed as follows. During slug migration, some slugs lose small parts of their back parts. In order to avoid tracking of these slug parts, objects that did not move more than $10^{-3}$ mm between two successive frames were removed from the list of slugs tracked.

Some slugs may cross each other along their trajectory, distorting slug ID tracking. Thus, to avoid any distortion of estimates, slugs with an area standard deviation of more than 5000 pixels were removed from the list of slugs.

To obtain best estimators of trajectories and morphological parameters, slugs that were not tracked on more than 20 frames and morphologically measured for less than 10 frames were removed. At the end of this data pre-processing, a total of 661 and 593 slugs were analysed for light and dark condition respectively.

## Statistical analysis

Statistical analysis was performed using the Python library stats-models, a *p-value p*<0.05 was considered as significant. We first considered replicates for each condition (light and dark) and computed tow-way ANOVA of *direction variability* in order to test for replicate effect. Without significant effect, we then combined all replicates (for each condition) in order to obtain equal intervals of cell number composed of up to 110 slugs (for light condition).

Significance of pairwise comparison of slug length and number of cells within slugs when exposed to light or when dark was established using the Wilcoxon test. Experimental data involving two independent variables (velocity, direction variability, linear time and Start end distance) were analysed using a two-way ANOVA. The two-way ANOVA was used to test for an interaction between these independent variables and the dependent variable (light, number of cells and number of cells*light).

Statistical directional analysis was performed using the Rayleigh z-test to test the null hypothesis that there is no sample mean direction. We first computed rectangular coordinates of $\theta_{SE}$ using following equations for a sample of size *n*:

$$X = \sum_{i=1}^{n} cos(\theta_{SE})/n$$
$$Y = \sum_{i=1}^{n} sin(\theta_{SE})/n \tag{12}$$

and *r*, the mean vector

$$r = \sqrt{X^2 + Y^2} \tag{13}$$

A $r$ value of 0 means uniform dispersion whereas a $r$ value of 1 means complete concentration in one direction.

We then determined the Rayleigh z statistic using the equation:

$$z = nr^2 \qquad (14)$$

Critical values of $z$ were taken from Table B.34, Zar (1999) [25] giving the *p-value*.

## Results

### Phototaxis is an emergent property of multicellularity in *D. discoideum*

Our first aim was to explore whether slug phototaxis results from phototaxis at the single-cell level or whether it is an emergent property of multicellularity. For this we developed an experimental setup and an image analysis pipeline to record and analyse migration trajectories of slugs and single cells in various conditions (Fig 1). Our data bridge the macroscale to track slug migration ($\approx 1$ mm in size over several cm) and the microscale to determine the number of cells within slugs and track single cell trajectories ($\approx 5$ microns in size over several tens of microns).

Directional statistical analysis were computed from Eq (13) for $r$ values and Rayleigh test for uniform angles distribution from Eq (14). A $r$ value of 0 means uniform dispersion whereas a $r$ value of 1 means complete concentration in one direction. Slugs exposed to lateral light turned and migrated towards the light source within 2 mn, and displayed significantly directional migration (directional analysis of start-end direction $\theta_{SE}$, from Eq (2): $r = 0.74$; $z_{365.2}$ p-value $<10^{-3}$). In contrast, in the dark, slug migration exhibited no preferential direction ($r = 0.067$; $z_{2.65}$ p-value $>0.05$) and showed spontaneous turning (Fig 2A combining all replicates, see also S2_FigA for each replicate). We then asked whether this multicellular property is already present at the single cell level. We prepared vegetative cells and cells extracted from dis-aggregated phototactic slugs, and monitored their migration either with lateral light or in the dark. Monitored cells were most likely at a low enough density to avoid cell-cell interactions. In contrast to slug migratory behavior, we did not observe any preferential direction for single cells when subjected to lateral light ($r = 0.018$; $z_{0.390}$ p-value $> 0.05$;

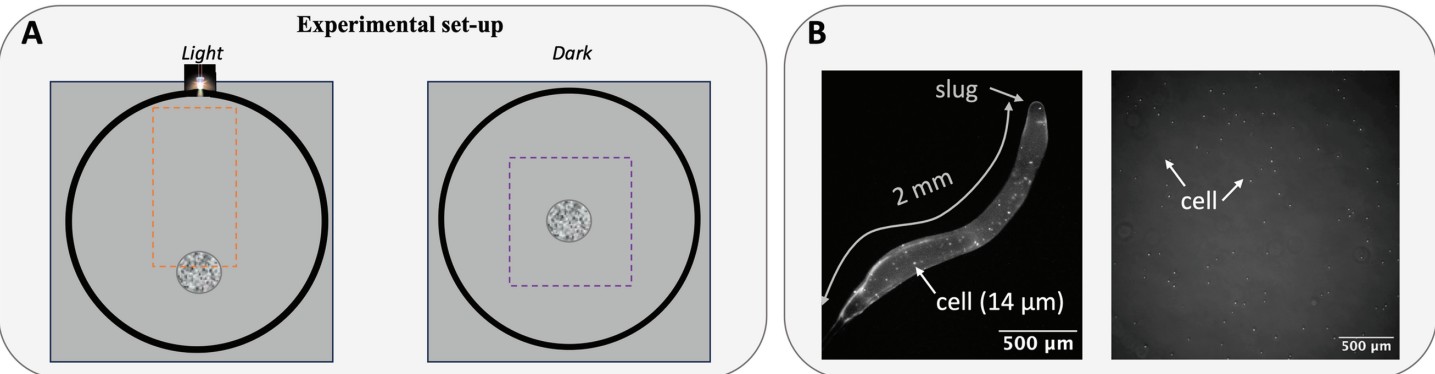

**Fig 1. Experimental set-up and slug versus cell size.** (**A**) Experimental set-up: on the left, cells were placed at 1 cm from the periphery of the Petri dish (gray ring) at the opposite side from the light source. On the right, cells in dark condition were inoculated into the center of the Petri dish. Dashed lines represent region of interest where slugs migrated. (**B**) Comparison between slug and cell size: fluorescence imaging of a slug composed of 1% fluorescent GFP cells (left) and vegetative cells (right).

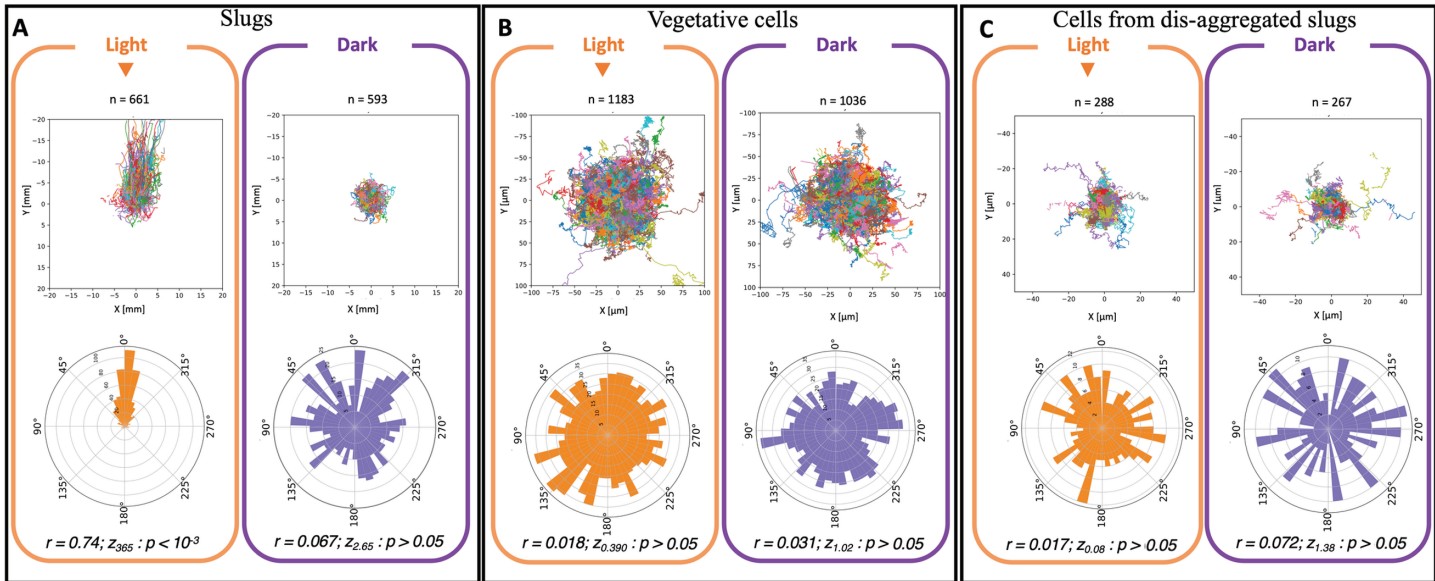

**Fig 2. Phototaxis is an emergent property of multicellularity.** Trajectories (centered from Eq (1)) and polar representations of start-end directions $\theta_{SE}$ (from Eq (2)) of slugs (**A**) and cells (**B** and **C**) when exposed to light (orange) or in the dark (violet). Slug migration showed no preferential direction in the dark but was oriented towards the light source under lateral illumination (A). Vegetative cells (B) and cells from dis-aggregated slugs (C) showed no preferential direction whether in the dark or exposed to lateral light. Directional statistical analysis were computed from Eq (13) for $r$ values and Rayleigh test for uniform angles distribution from Eq (14).

Fig 2 B for vegetative cells and Fig 2 C for cells from dis-aggregated phototactic slugs: $r = 0.017$; $z_{0.08}$ *p-value* > 0.05 , see also S2_Fig B and C for each replicate).

This result was consistent with other works [13–15] and suggests that phototaxis is a collective property that is absent at the single cell level. This hypothesis was also supported by an increased multicellular velocity (10.5 µm/min $\pm$ 0.2) compared to single cell velocity of dis-aggregated slugs (0.59 µm/min $\pm$ 0.01) (Student t-test: p-value $<10^{-3}$).

We repeated the same experiment at a higher cell density to explore whether phototaxis could emerge from cell-cell interactions but without the multicellular organization present at the slug stage. As for cells at low density, no phototaxis was detected (see S2_FigD) suggesting that self-organization was needed for phototaxis to emerge.

## Dependence of phototactic and migration behaviours with slug size

In the second part of this work, we explored how slug size impacts phototaxis and the transition of this emergent property. We monitored the migration of a large number of slugs under lateral light or in the dark and developed computational image analysis at the macro-scale and at the micro-scale to quantify slug migratory behaviour as a function of slug size over a wide range.

**Measuring the number of cells within living slugs** To measure the number of cells composing slugs, we formed slugs containing 1% of red fluorescent reporter cells among 99% green fluorescent cells, counted the number of red fluorescent cells in each slug at the microscale, and deduced the total number of cells in each slug (as cell division ceases during multicellular development). At 1% red cells are well spatially separated, allowing to count them in each slug using standard fluorescence microscopy. The reciprocal mix of green and red cells was also performed to control for fluorescence reporter biases. To avoid interfering with

our phototaxis assay upon exposure with intense excitation light during fluorescence imaging, we then calibrated slug length vs number of cells composing a slug. We noticed that slugs with a given number of cells were longer when exposed to lateral light (S3_FigA and S3_FigB) (Wilcoxon test light versus dark: length p-value $<10^{-3}$; ANOVA length * number of cells p-value = 0.0182). We thus performed two separate calibrations for slugs in the dark vs exposed to lateral light (see S3_FigC and S3_FigD). The linear correlation we found between slug length and number of cells in each case allowed us to quantify the number of cells composing slugs during our phototaxis assay by extracting slug length upon phase contrast imaging, which entails a much less intense light exposure than fluorescence acquisition.

**Phototaxis efficiency increases gradually with cell density** We analysed 661 slug trajectories when exposed to lateral light and 593 slug trajectories when in the dark, by combining 3 to 4 replicates with 2 sub-replicates (see S1_Table). This allowed us to obtain a large range of slug sizes. Combining all replicates was possible after analysing each replicate for dark and light conditions and observing no significant difference between replicates (see S4_Fig for analysis of direction variability between replicates from Eq (4)).

In order to explore the dependence of *D. discoideum* phototaxis with slug size, we quantified globally (Fig 3A-a) and locally (Fig 3A-b) the directionality of slug trajectories, in the dark or under lateral illumination, over a wide range of slug sizes. Namely, we estimated start-end directions ($\theta_{SE}$) and instantaneous directions relative to the direction of the light source (following Eq (2) and Eq (3) respectively).

Start-end directions (Fig 3B-a) are widely and homogeneously distributed, showing no preferred direction for slugs in the dark regardless of their size, and for slugs with less than $2.1 \times 10^3$ cells under lateral light. Slugs with more than $2.1 \times 10^3$ cells show a preferred direction coinciding with the light direction, with a bias towards the light source that increases continuously with slug size.

Instantaneous directions (Fig 3B-b) display a similar pattern. The alignment of instantaneous directions with the light source direction increases with slug size under lateral illumination, with a weakly detectable preferred orientation aligned with the light source direction for the smallest slugs with less than $2.1 \times 10^3$ cells. For slugs in the dark, there is no detectable preferred instantaneous direction, regardless of slug size.

Accordingly, estimation of angular distribution parameters shows that start-end directions are essentially random for slugs in the dark regardless of their size and small slugs under lateral illumination with $\kappa \approx 0$ (estimated from Eq (5)), while the start-end directional bias continuously increases with slug size to reach $\kappa \approx 8$ for the largest slugs (Fig 3C-a). For instantaneous directions, the trend is similar except that $\kappa$ increases from $\approx 0.5$ to $\approx 2$ with increasing slug size under lateral illumination, while it remains below $\approx 0.2$ for slugs in the dark at all slug sizes (Fig 3C-b). For polar representations of each category, see also S5_Fig. Phototaxis efficiency, defined as the $\kappa$ parameter of von Mises-fitted orientation distributions, significantly increases with increasing slug size (Fig 3C-a and 3C-b, ANOVA: number of cells: p-value = 0.017 and 0.71 for light and dark condition respectively; ANOVA dark vs light: number of cells: p-value = 4.9 $10^{-3}$, light: p-value = 2.8 $10^{-4}$, number of cells*light: p-value = 4.2 $10^{-3}$ for kappa ($\theta_{SE}$)).

We conducted several other types of trajectory analyses. Direction variability (Eq (4), corresponding to the variance of instantaneous direction S6_FigB), linear time (the time during which the slug's changing direction is lower than 10 degrees between two successive time steps, S7_FigA-a), decomposition of the velocity vector into components parallel and orthogonal to the light source direction (Eq (8), S7_FigA-b and S7_FigA-c) and start-end distance (Eq (10), S7_FigA-d). All of these analyses demonstrated continuously increasing phototaxis

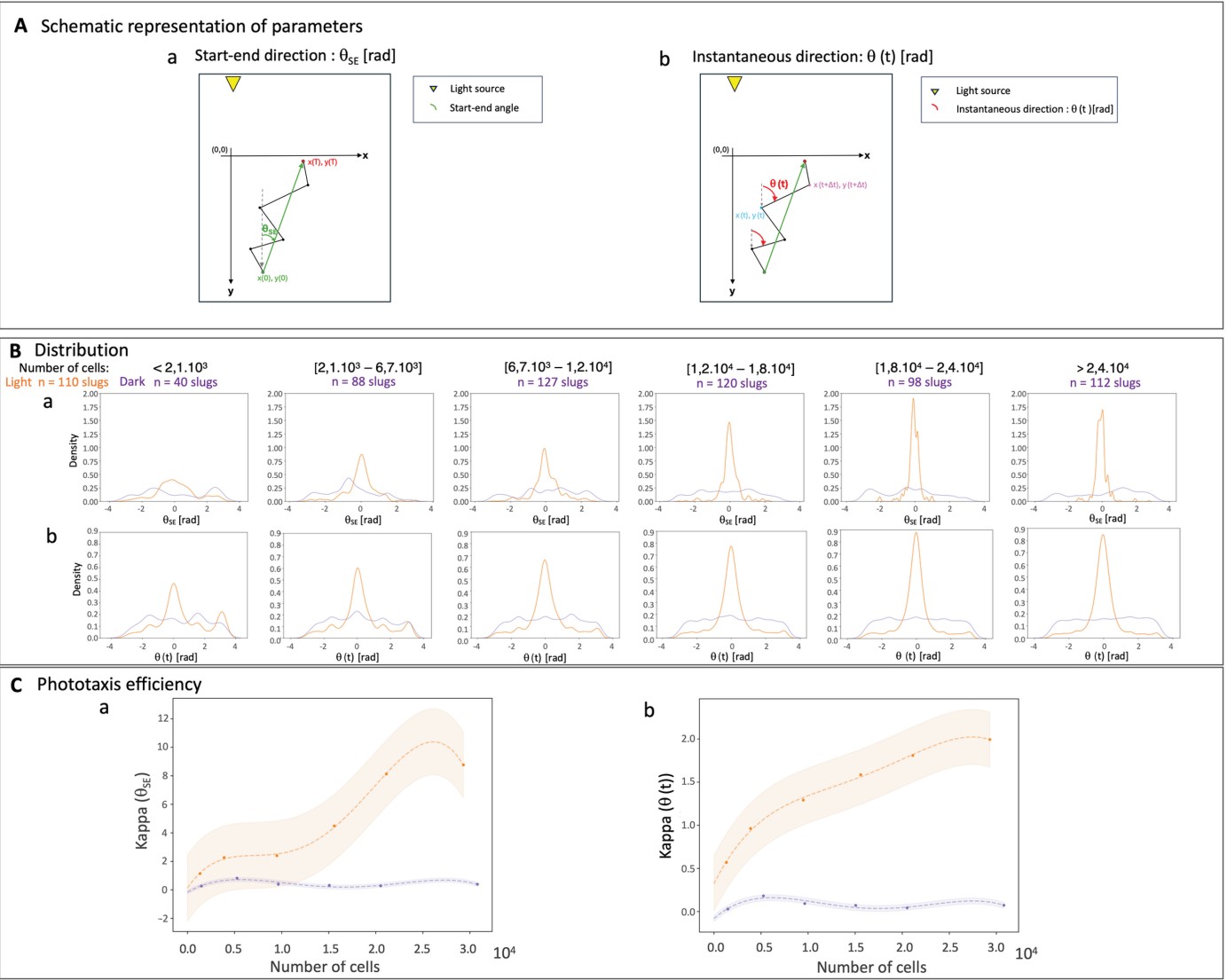

**Fig 3. Phototaxis efficiency as a function of slug size**: (**A**) Schematic representation of parameters: start-end direction estimated from Eq (2) (a) and instantaneous direction from Eq (3) (b). Distribution of start-end directions (B-a) and distribution of instantaneous directions relative to the light source direction (**B**-b) for 6 intervals of slug sizes (cell number) under lateral illumination (orange) or in the dark (violet). Corresponding phototaxis efficiency (**C**) estimated from Eq (5). Points in C were fitted with a polynomial function of degree 4. Phototaxis efficiency increases gradually with slug size.

as a function of slug size. These analyses are thus all consistent with the global/local direction analyses.

## Discussion

Transitions to multicellularity have often been accompanied by the emergence of new properties that may confer advantages over ancestral single cells. Many self-organizing systems exhibit emergent behaviours.

In our study, results at the single cell level (vegetative cells and dis-aggregated slugs) combined with observations at the higher level of organization (multicellular stage) of the same biological system revealed that phototaxis of the social amoeba *Dictyostelium discoideum*

is an emergent property of multicellularity. Phototaxis emerged from interactions between individual cells at the higher level of organization within slugs. Our results are in agreement with other works demonstrating no phototaxis at the single cell level [13–15] but efficient migration towards light at the multicellular stage [26]. Individual vegetative cells at high density (without self-organization), individual cells isolated from phototactic slugs and small slugs (with self-organization but too few cells) did not respond to light suggesting that both interactions between cells and self-organization was a prerequisite for this new property to emerge at the collective level. Single cells from phototactic slugs were assayed 30 mn following slug dis-aggregation (the time for cells to attach to the surface), which is shorter than typical timescales of transcriptional [27] and *a fortiori* proteome changes in *Dictyostelium*, with the exception of post-translational modifications. We thus expect single cells from disaggregated slugs to have very similar gene expression profiles as cells in phototactic slugs, in particular for phototaxis-required genes. We thus conclude that interactions between cells are key for phototaxis.

How cell-cell interactions control phototaxis remains an open question. These interactions may be required for either light detection, response to light, or coordination of cell motion. Several genes (including important signalling regulators) have been reported to be essential for phototaxis [28,29]. Within slugs, cell-cell interaction such as communication and signalling may play a crucial role indeed for the emergence of this property [30]. For small slugs a decreased communication between cells could explain a decrease in phototaxis efficiency. Indeed, Bonner et al [31] proposed that light-induced ammonia production causing cell movement at the slug tip accumulated to a limited extent in small slugs due to ammonia dissipation.

Phototaxis efficiency in *D.discoideum* increases with the number of cells within slugs (Fig 3). In contrast to the size-dependent threshold reported for sensory response of ant colonies [11], we find that phototaxis efficiency increases continuously with group size (Fig 3 and S7_Fig).

The size dependence was also observed for slug speed with larger slugs moving faster and more linearly compared to smaller slugs (S6_Fig). Our result was also in agreement with other works showing faster migration for larger slugs [12,32,33]. Other experimental studies have emphasized a crucial role of regulating speed in collective behaviours [34,35] and a relationship between ordered states, local population size and speed [36]. This suggests an essential association between speed and the degree to which individuals coordinate their motion. Coordinated cell motion within slugs likely promotes the cohesive behaviour of cells as a unique entity that is highly responsive to environmental information. However, transient environmental information at the individual cell level may cause a delay in the collective integration of these external signals. In other Amoebozoa species, this process has been found to result in a trade-off between signal orientation accuracy and migration speed, thought to be a fundamental feature of biological information processing [37,38].

We now discuss possible evolutionary scenario of the emergence of phototaxis in amoebozoa. The phototactic behaviour in Dictyostelium slugs can be driven by selective pressures favouring effective response to light and is believed to be beneficial. Indeed phototaxis allows slugs to reach the soil surface, thus increasing spore dispersion efficiency. This size-dependent phototaxis-driven advantage enhances the selection for large slug sizes that enhances spore production [39]. Body size, influenced by ecological factors like habitat, climate, and biotic interactions, plays a crucial role in evolutionary adaptation and should act on the gain or loss of some Dictyostelia features, providing the ultimate cause for phenotypic innovation [40]. Phototactic behaviour is strongly correlated with size across and within species [33]. Despite limited wild data and sampling, phylogenetic and phenotypic analyses showed that in

Dictyostelia, large structures, phototropism and slug migration coevolved as evolutionary innovations [40].

## Author contributions

**Conceptualization:** Clément Nizak, Sandrine Adiba.

**Data curation:** Damien Genettais, Clément Nizak, Sandrine Adiba.

**Formal analysis:** Damien Genettais, Clément Nizak, Sandrine Adiba.

**Funding acquisition:** Sandrine Adiba.

**Investigation:** Damien Genettais, Charles Bernard, Félix Geoffroy, Clément Nizak, Sandrine Adiba.

**Methodology:** Clément Nizak, Sandrine Adiba.

**Software:** Damien Genettais, Sandrine Adiba.

**Supervision:** Clément Nizak, Sandrine Adiba.

**Validation:** Damien Genettais, Clément Nizak, Sandrine Adiba.

**Visualization:** Damien Genettais, Clément Nizak, Sandrine Adiba.

**Writing – original draft:** Damien Genettais, Clément Nizak, Sandrine Adiba.

## Supporting information

**Table 1. Cell densities under light and dark conditions.** Cell densities are presented as the number of cells per square centimeter ($cm^2$). Light and dark conditions were tested across multiple replicates and sub-replicates.
(PDF)

**S1 Fig. Experimental Set-up and analysis**. (**A**) Macroscale and microscale image analysis. a: Time-lapse acquisition was performed every 10 minutes for 50 hours using a 5X objective and phase contrast. b: Macroscopic reconstruction (1) from images taken at the microscale. The orange dashed rectangle corresponded to the Region Of Interest where slugs migrated. The red dashed polygon represented the area of cell aggregation removed from the analysis. (c) Slugs were tracked with wrMTrck imageJ plugin (2) yielding slug trajectories, (x,y) coordinates. From these coordinates, we obtained corresponding columns, raws, and times to analyze microscale images and extract morphological data. d: Morphological analysis of images (3) at the microscale (dash yellow square in a). Slugs that crossed the border were not analysed. The corresponding slug ID at the microscale ($ID_\mu$) was assigned to the slug ID tracked at the macroscale ($ID_M$). Image acquisition at the microscale (a), macroscopic reconstruction (b) combined with analysis (1,2,3) were used to obtained phototaxis data. e: Morphological and cell number within slugs analysis (3b) performed at the microscale. This experiment was performed using phase contrast and fluorescent images in order to determine the cell number within living slugs. Phase contrast channel (left), fluorescent channel (center), and corresponding local maxima (right) detected using local maxima detection from the skiimage Python library. (**B**) Trajectories obtained from slug tracking with the wrmtrack imageJ plugin in light (left) or dark (right) conditions within Regions of Interest, after removing the area of cell aggregation (red hashed polygons). All experiments were analysed by selecting Region

Of Interest including slugs trajectories (orange and violet dashed rectangles for light and dark condition respectively).
(TIFF)

**S2 Fig. Individual trajectories (upper plots) according to Eq (1) and polar distributions of directions (from Eq (2)) (lower plots) in lateral light (orange) and dark (violet) conditions**. Slugs moved towards the light source (**A**). Cells from dis-aggregated slugs (**B**), vegetative cells at low density (**C**), and cells plated at high density (**D**) and exposed to light did not display any phototactic behaviour.
(TIFF)

**S3 Fig. Number of cells within slugs as a function of slug length**. (**A**) Distribution of slug lengths when exposed to lateral light or in the dark. Slugs migrating towards lateral light exhibit a wider length range compared to slugs migrating in the dark. When slugs were exposed to lateral light, we observed significantly higher mean slug length compared to slug migration in the dark (0.90 mm $\pm$0.32 mm and 0.71 mm $\pm$0.22 mm in light and dark conditions respectively, Wilcoxon test *p-value*<$10^{-4}$). Light induced slug elongation such that slugs with the same number of cells (**B**) were longer when exposed to light (ANOVA length * number of cells *p-value*=0.0182). (**C**) Slug length correlation when slugs were exposed to lateral light (magenta): cell number = 34 252 * length - 10 681 and in the dark (red): cell number = 51 085 * length - 10 749. For small length values, the correlation was fitted using a polynomial equation (dashed lines), for light: cell number = 20 409 (length)$^2$ ; for dark: cell number = 41 194 (length)$^2$. (**D**) Slug length correlates with the number of cells whatever the fluorescent marker (GFP in green or RFP in red) Each point corresponds to one slug, lines correspond to linear regression, dashed lines to polynomial fit.
(TIFF)

**S4 Fig. Direction variability for each replicate and sub-replicate**. When slugs migrated towards light (**A**) direction variability decreased significantly with the increased number of cells (p-value<$10^{-3}$) within slugs for all replicates. When in the dark (**B**), direction variability displayed no variation (p-value>0.05) with the number of cells within slugs. No significant difference was observed between replicates and sub-replicates (ANOVA number of cells*replicate-sub-replicate, light: *p-value*=0.48; dark: p-value=0.55).
(TIFF)

**S5 Fig. Polar distribution of slug directions for various slug size ranges (number of cells per slug).** Each interval was composed of an equal and sufficient number of slugs for statistical analysis. Large slugs exhibit directional migration when exposed to light (upper line, orange), compared to dark conditions (violet). The *r* value computed from Eq (13) ranged from 0.046 to 0.21 and 0.47 to 0.91 for dark and light conditions respectively (from lower to higher slug sizes).
(TIFF)

**S6 Fig.** Analysis of slug speed showed strong dependence on the cell number whatever illumination conditions (with light or in the dark, ANOVA number of cells: *p-value*<$10^{-4}$). We observed no significant differences on slug speed between dark and light conditions (0.38 mm/h $\pm$0.007 mm/h and 0.63 mm/h $\pm$0.01 mm/h for dark and light conditions respectively, ANOVA number of cells: *p-value*<$10^{-4}$, light: *p-value*<$10^{-4}$, number of cells * light: *p-value*=0.8). Therefore, the slug speed was independent of the presence of the light but dependent of the number of cells within slugs. Previous works studying the light effect on slug speed have yielded conflicting results. Some authors reported an increase in slug speed following light irradiation [12,41,42] whereas others reported no changes on slug speed upon

light [13,43]. (**A**) Slug velocity for slugs exposed to light or in the dark. Slug velocity increased with the number of cells whatever the illumination condition (light: orange or dark: violet) and exhibited same slope whatever the illumination condition. (**B**) Instantaneous direction variability (calculated following Eq (4)) as a function of cell number within slugs when in the dark (violet) and exposed to light (orange). Larger slugs exhibited a more linear trajectory when exposed to lateral light, but not in the dark. (**C**) Distribution of instantaneous directions for slugs exposed to light (orange) or in the dark (violet). Corresponding *kappa* values quantifying bias of migration (following Eqs (2) and (5)).
(TIFF)

**S7 Fig.** Linear time (**A**-a), parallel and orthogonal instantaneous velocity components (A-b for light and c for dark, from Eq (8)) and start-end distance (A-d computed from Eq (10)). All of these analyses showed continuously increasing phototaxis with slug size. In (**B**) examples of slug trajectories for each category of slug sizes. Slugs with small number of cells migrated at the same distance compared to slugs in the dark. Slugs migrating in the dark did not explore very far, whatever the number of cells inside slugs (violet) (for the same ranges as Fig 3).
(TIFF)

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
