## [Decision Letter · Decision Letter 0]

13 Jan 2025

PONE-D-24-44222Continuous emergence of phototaxis in Dictyostelium discoideumPLOS ONE

Dear Dr. Adiba,

Thank you for submitting your manuscript to PLOS ONE. After careful consideration, we feel that it has merit but does not fully meet PLOS ONE’s publication criteria as it currently stands. Therefore, we invite you to submit a revised version of the manuscript that addresses the points raised during the review process.

We look forward to receiving your revised manuscript.

Kind regards,

Michael Klymkowsky, Ph.D.

Academic Editor

PLOS ONE

2. Thank you for stating the following financial disclosure:  [This work has received support under the program "Investissements d’Avenir" launched by the French Government and implemented by ANR with the references ANR–10–LABX–54 MEMOLIFE and ANR–10–IDEX–0001–02 PSL* Université Paris, Q-life ANR-17-CONV-6150005.].  Please state what role the funders took in the study.  If the funders had no role, please state: "The funders had no role in study design, data collection and analysis, decision to publish, or preparation of the manuscript." If this statement is not correct you must amend it as needed. Please respond by return e-mail so that we can amend your financial disclosure and competing interests on your behalf.

3. We notice that your supplementary figures are included in the manuscript file. Please remove them and upload them with the file type 'Supporting Information'. Please ensure that each Supporting Information file has a legend listed in the manuscript after the references list.

Additional Editor Comments:

Please note that I have acted as a reviewer for this manuscript, and you will find my comments below, under Reviewer 2.

Reviewers' comments:

Reviewer's Responses to Questions

**Comments to the Author**

1. Is the manuscript technically sound, and do the data support the conclusions?

Reviewer #1: Yes

Reviewer #2: Yes

2. Has the statistical analysis been performed appropriately and rigorously? 

Reviewer #1: Yes

Reviewer #2: Yes

3. Have the authors made all data underlying the findings in their manuscript fully available?

Reviewer #1: Yes

Reviewer #2: Yes

4. Is the manuscript presented in an intelligible fashion and written in standard English?

Reviewer #1: Yes

Reviewer #2: Yes

5. Review Comments to the Author

Reviewer #1: The manuscript by Genettais et al. describes results of experiments that are technically sound and data that clearly supports the conclusions. The statistical analysis appears to be appropriate. The manuscript is well-written. I have a few minor suggestions for improvement:

• In the Introduction (last sentence of the third paragraph), I suggest adding more explanation of the contrasting observations. This sentence is too vague. I would like to see more discussion on previous studies which have asked the same question. Without a discussion on the contrasting observations, one of their main conclusions that individual cells do not exhibit phototaxis is less novel or noteworthy.

• The authors use a “slugger” mutant (mybC-) for their studies. I suggest adding more discussion of why the authors do not think this mutation would interfere with individual cell phototaxis.

• I suggest making the font bigger in Figure 3 and b.

Reviewer #2: The manuscript by Genettais et al., on the emergence of phototactic behavior in multicellular Dictyostelium slugs is straightforward and elegant. I concur with reviewer #1 that the conclusions appear well supported and the statistically analyses are sound (although statistics is not my strongest area of expertise).

My one concern, and this is something for follow-up studies, is that the assumption that changes in gene expression/protein function are not involved in the transition from multicellular to single cell behavior (upon slug dissociation) may be too sweeping. It is possible that this transition may activate various cellular stress response systems, including changes to gene expression / protein activity, etc.

It will be extremely interesting to see what changes occur at the single cell level over time, but again that is clearly for later.

Very minor points. on page 2 of the manuscript, I would replace "large amount" with "large number " of slug trajectories and I believe the appropriate abbreviation for minutes is min rather than "nm".

6. PLOS authors have the option to publish the peer review history of their article (what does this mean?). If published, this will include your full peer review and any attached files.

Reviewer #1: No

Reviewer #2: No

---

## [Author Response · Author response to Decision Letter 1]

20 Feb 2025

Dear Editor,

We thank you for your positive assessment and thoughtful comments.

We provide here a brief summary of the main changes (marked in blue in the 'Revised Manuscript with track changes'), and proceed later to answer comments point-by-point.

1. We have revised our manuscript to ensure it adheres to PLOS ONE's style requirements. The manuscript now aligns with the provided templates, main text sections, figures, and tables. File names also follow the naming conventions specified by PLOS ONE.

2. We confirm that the following financial disclosure is accurate:

“This work has received support under the program ‘Investissements d’Avenir’ launched by the French Government and implemented by ANR with the references ANR–10–LABX–54 MEMOLIFE and ANR–10–IDEX–0001–02 PSL Université Paris, Q-life ANR-17-CONV-6150005.”

Additionally, we have included the statement (line 450): "The funders had no role in study design, data collection and analysis, decision to publish, or preparation of the manuscript."

If any further modifications are required, please let us know

3. We have removed the supplementary figures and files from the main manuscript and uploaded them separately as ‘Supporting Information’ files. Each supplementary file includes a legend, which has also been added to the manuscript after the references list, in compliance with PLOS ONE's guidelines

4. We have carefully reviewed our reference list to ensure that it is complete and accurate

Thank you for considering our revised manuscript for publication. We look forward to your feedback and hope for the opportunity to contribute to your journal.

Please find the revised attached files.

Reviewer 1:

In the Introduction (last sentence of the third paragraph), I suggest adding more explanation of the contrasting observations. This sentence is too vague. I would like to see more discussion on previous studies which have asked the same question. Without a discussion on the contrasting observations, one of their main conclusions that individual cells do not exhibit phototaxis is less novel or noteworthy.

We agree that the summary of previous studies at the end of the introduction was too vague. In short, prior studies followed different experimental approaches, probably explaining part of the discrepancies between their respective conclusions. These approaches were also probing motion indirectly, impeding one to draw quantitative conclusions at the single-cell and multi-cellular scales. Our new approach aims precisely at analyzing motion directly at both scales. To make this point clearer we have added the following sentence at the end of the third paragraph in the introduction (line 30-35):

"On the other hand, phototaxis, that has been widely reported in multicellular slugs, has led to contrasting observations concerning the response of isolated cells. Some studies [1-3] report no evidence of phototaxis in single cells, while others [4-6] show cell accumulation or dispersion near a light source. These discrepancies may stem from the different experimental methodologies used in these studies. In addition, these methodologies characterized cell or slug motion mainly indirectly, e.g. via the analysis of traces left behind by cells or slugs on agar. To revisit this question, we have thus developed an approach for the direct visualization and statistical analysis of single-cell and slug motion in controlled lateral illumination conditions."

The authors use a 'slugger' mutant (mybC-) for their studies. I suggest adding more discussion of why the authors do not think this mutation would interfere with individual cell phototaxis.

We agree that our justification for using a mybC- mutant was too weak. We specifically used the mybC- mutant because it provides a unique advantage: it forms slugs capable of migrating for an extended period. This makes it particularly suitable for studying phototaxis at the multicellular level, as it allows prolonged and stable observations of light-driven behavior, and thus larger samples for statistical analysis of slug motion. Our pilot experiments at the onset of our project indicated that our choice does not affect our conclusions on phototaxis efficiency depending on slug size and being undetectable in single cells. Consistent with this, we now cite a paper showing that the mybC mutation does not impact the organization of the multicellular slug but instead affects culmination, a later stage of the developmental cycle.

While our current work focuses on phototaxis mechanisms in these contexts, a more detailed exploration of how the mybC mutation might influence specific sensory or signaling pathways could be an interesting study in its own right.

We added the sentence line 57-59 in material and methods:

"The mybC mutation was reported not to impact the organization of the multicellular slug but instead to affect culmination, a later stage of the developmental cycle [7].”

I suggest making the font bigger in Figure 3 and b.

We increased the font size in Figure 3 and b.

Reviewer 2:

My one concern, and this is something for follow-up studies, is that the assumption that changes in gene expression / protein function are not involved in the transition from multicellular to single cell behavior (upon slug dissociation) may be too sweeping. It is possible that this transition may activate various cellular stress response systems, including changes to gene expression or protein activity, etc.

It will be extremely interesting to see what changes occur at the single cell level over time, but again that is clearly for later.

We acknowledge that this point was not sufficiently discussed in our original manuscript. Nichols et al [8] analyzed transcriptional changes during differentiation and de-differentiation, the latter being triggered by feeding cells from disaggregated slugs. Their time-course analysis demonstrated that transcriptional changes occur in both cases over a typical timescale of several hours, with the fastest gene expression changes taking about 1h. At the protein level, changes must be a fortiori slower, except for post-translational modifications. We thus argue that by and large, genes expressed in phototactic slugs are still expressed in cells from disaggregated phototactic slugs that we observed 30 mn following disaggregation, in particular genes required for phototaxis. Accordingly, we have added the following sentence in our discussion (line 394-400):

"Single cells from phototactic slugs were assayed 30 mn following slug dis-aggregation (the time for cells to attach to the surface), which is shorter than typical timescales of transcriptional [8] and a fortiori proteome changes in Dictyostelium, with the exception of post-translational modifications. We thus expect single cells from disaggregated slugs to have very similar gene expression profiles as cells in phototactic slugs, in particular for phototaxis-required genes. We thus conclude that interactions between cells are key for phototaxis."

However, we agree that we cannot exclude that disaggregation involves more complex cellular mechanisms, including stress responses with exceptionally fast changes in gene expression and protein activity. Single-cell RNA sequencing could provide insights into whether specific stress response pathways are activated at the single-cell level. Exploring these questions in depth would require dedicated experiments and analysis, making it an important and valuable study.

Very minor points. on page 2 of the manuscript, I would replace "large amount" with "large number " of slug trajectories and I believe the appropriate abbreviation for minutes is min rather than "nm".

We appreciate your suggestion to replace "large amount" with "large number" of slug trajectories. This change has been implemented for accuracy line 47.

Regarding the abbreviation "mn," we would like to clarify that we have indeed used "mn" consistently to indicate minutes throughout the manuscript, in alignment with standard conventions.

As for the term "nm" on line 117, we confirm that this refers to nanometers, representing a wavelength, which is the conventional abbreviation for this unit in scientific literature.

With best regards,

Clément Nizak and Sandrine Adiba.

1. Bonner JT, Whitfield FE. The relation of sorocarp size to phototaxis in the cellular slime mold, Dictyostelium purpureum. The Biological Bulletin. 1965;128(1):51–57.

2. Francis DW. Some studies on phototaxis of Dictyostelium. Journal of Cellular and Comparative Physiology. 1964;64(1):131–138.

3. Samuel EW. Orientation and rate of locomotion of individual amebas in the life cycle of the cellular slime mold Dictyostelium mucoroides. Developmental biology. 1961;3(3):317–335.

4. Häder DP, Poff KL. Light-induced accumulations of Dictyostelium discoideum amoebae; 1979.

5. Häder DP, Vollertsen B. Phototactic orientation in Dictyostelium discoideum amoebae. Acta protozoologica. 1991;30(1).

6. Häder DP, Claviez M, Merkl R, Gerisch G. Responses of Dictyostelium discoideum amoebae to local stimulation by light. Cell biology international reports. 1983;7(8):611–616.

7. Guo K, Anjard C, Harwood A, Kim HJ, Newell PC, Gross JD. A myb-related protein required for culmination in Dictyostelium. Development (Cambridge, England). 1999;126(12):2813 – 2822.

8. Nichols JM, Antolovic V, Reich JD, Brameyer S, Paschke P, Chubb JR. Cell and molecular transitions during efficient dedifferentiation. Elife. 2020;9:e55435

---

## [Decision Letter · Decision Letter 1]

9 Mar 2025

Continuous emergence of phototaxis in Dictyostelium discoideum

PONE-D-24-44222R1

Dear Dr. Adiba,

We’re pleased to inform you that your manuscript has been judged scientifically suitable for publication and will be formally accepted for publication once it meets all outstanding technical requirements.

Kind regards,

Dave Mangindaan

Academic Editor

PLOS ONE

Additional Editor Comments (optional):

Please address the suggestion from Reviewer #2. This can be performed during the reading of galley proof stage.Thank you.

Reviewers' comments:

Reviewer's Responses to Questions

**Comments to the Author**

1. If the authors have adequately addressed your comments raised in a previous round of review and you feel that this manuscript is now acceptable for publication, you may indicate that here to bypass the “Comments to the Author” section, enter your conflict of interest statement in the “Confidential to Editor” section, and submit your "Accept" recommendation.

Reviewer #1: All comments have been addressed

Reviewer #2: All comments have been addressed

2. Is the manuscript technically sound, and do the data support the conclusions?

Reviewer #1: Yes

Reviewer #2: Yes

3. Has the statistical analysis been performed appropriately and rigorously? 

Reviewer #1: Yes

Reviewer #2: I Don't Know

4. Have the authors made all data underlying the findings in their manuscript fully available?

Reviewer #1: Yes

Reviewer #2: Yes

5. Is the manuscript presented in an intelligible fashion and written in standard English?

Reviewer #1: Yes

Reviewer #2: Yes

6. Review Comments to the Author

Reviewer #1: (No Response)

Reviewer #2: I would recommend removing "merely" from the abstract

There are some awkward (in english) phrasing here and there that might be readily identified by asking ChatGPT or Claude to find and fix them (just a suggestions).

7. PLOS authors have the option to publish the peer review history of their article (what does this mean?). If published, this will include your full peer review and any attached files.

Reviewer #1: No

Reviewer #2: **Yes: **Michael Klymkowsky

---

## [Editor Report · Acceptance letter]

PONE-D-24-44222R1

PLOS ONE

Dear Dr. Adiba,

I'm pleased to inform you that your manuscript has been deemed suitable for publication in PLOS ONE. Congratulations! Your manuscript is now being handed over to our production team.

Kind regards,

on behalf of

Assoc. Prof. Dave Mangindaan

Academic Editor

PLOS ONE